# Safety and Efficacy of Regadenoson for Pediatric Stress Perfusion Cardiac MRI with Quantification of Myocardial Blood Flow

**DOI:** 10.3390/children9091332

**Published:** 2022-09-01

**Authors:** Shivani G. Patel, Nazia Husain, Cynthia K. Rigsby, Joshua D. Robinson

**Affiliations:** 1Division of Cardiology, Ann and Robert H. Lurie Children’s Hospital of Chicago, Chicago, IL 60611, USA; 2Department of Pediatrics, Feinberg School of Medicine, Northwestern University, Chicago, IL 60611, USA; 3Department of Medical Imaging, Ann and Robert H. Lurie Children’s Hospital of Chicago, Chicago, IL 60611, USA; 4Department of Radiology, Feinberg School of Medicine, Northwestern University, Chicago, IL 60611, USA

**Keywords:** pediatric, stress perfusion MRI, regadenoson

## Abstract

Myocardial stress perfusion magnetic resonance imaging is a non-invasive tool to assess for myocardial ischemia and viability. Pediatric myocardial stress perfusion MRI can be challenging due to multiple intravenous lines, sedation, inadequate breath holding, fast heart rates, and complex anatomy. We performed a retrospective analysis in 39 children to evaluate safety and efficacy of regadenoson, a coronary vasodilator administered via a single intravenous line (6–10 mcg/kg), with respiratory motion correction (MOCO) and semi-quantitative blood flow analysis. Stress response data and adverse events were recorded, and image quality compared between native and MOCO reconstructions, assessing for perfusion deficits. Semi-quantitative analysis compared myocardial perfusion reserve index (MPRI) between patients who had a focal perfusion defect, patients who had undergone an orthotopic heart transplant, and non-transplant patients with no focal defects. Stress perfusion was completed in 38/39 patients (median age 15 years with a 41 ± 27% rise in heart rate (*p* < 0.005). Fifteen out of thirty-eight had transient minor side effects with no major adverse events. MOCO image quality was better than non-MOCO (4.63 vs. 4.01 at rest, *p* < 0.005: 4.41 vs. 3.84 at stress, *p* < 0.005). Reversible perfusion defects were seen in 4/38 patients with lower segmental mean MPRI in the area of the perfusion defect, nearing statistical significance when compared to non-transplant patients with no defects (0.78 ± 0.22 vs. 0.99 ± 0.36, *p* = 0.07). The global MPRI of the 16 patients who had undergone orthotopic heart transplant was significantly lower than the non-transplant patients (0.75 ± 0.22 vs. 0.92 ± 0.23, *p* = 0.03). Regadenoson is a safe and effective coronary vasodilator for pediatric stress perfusion MRI with MOCO producing better image quality and allowing for semi-quantitative assessment of perfusion deficits that correlate with qualitative assessment.

## 1. Introduction

Stress perfusion cardiac MRI is used as a non-invasive tool for qualitative and quantitative evaluation of myocardial blood flow through the coronary microcirculation and has been established in the diagnosis and risk stratification of coronary artery disease in adults [1,2,3,4,5]. Although nuclear medicine studies such as positron emission tomography (PET) are considered the gold standard in the assessment of myocardial viability, stress perfusion MRI has been shown to be an equally, if not more, sensitive and specific test for diagnosing coronary artery disease [1,2]. In addition, MRI can evaluate complex cardiac anatomical structure, myocardial structure, and function on the same test without the potentially harmful effects of ionizing radiation [6] making it a very useful tool in the pediatric population. In children, stress perfusion MRI has been used for evaluation of a broad array of disease processes, including congenital coronary artery anomalies, cardiac transplant surveillance, Kawasaki disease, and post-operative assessment of coronary artery patency [6,7,8,9,10]. Testing in pediatrics presents a unique set of challenges including technical difficulties and discomfort from intravenous line placement, greater need for sedation and lack of ability to cooperate with breath holding instructions. Moreover, the fast heart rates, complex anatomy, and artifact from motion and breathing tend to complicate image acquisition.

Adenosine, a coronary vasodilator, is the pharmacologic agent most used for stress perfusion MRI [11]. Challenges to its use in the pediatric population include the need for two intravenous (IV) lines due to its short half-life and the occurrence of side effects due to non-specific adenosine receptor activation (including hypotension, AV block and bronchospasm). Regadenoson is a vasodilator with selective cardiac A2A adenosine receptor activity and fewer side effects [12,13]. Its pharmacokinetics allow for a simplified fixed dose single bolus injection through a single peripheral IV cannula. Previous pediatric studies have shown the safety of regadenoson at a dose of 8–10 mcg/kg with a maximum dose of 0.4 mg [13,14,15]

Lastly, respiratory motion correction techniques have been shown to reduce exposure to anesthesia in children undergoing cardiac MRI [16] and have significantly improved stress perfusion image quality in adults [17]. Motion corrected (MOCO) stress perfusion MRI can add value to qualitative analysis of stress perfusion images by correcting for motion artifacts and allow improved semi-quantitative and quantitative assessment of myocardial perfusion. Quantification improves inter-reader reliability in diagnosing ischemia and grading severity of epicardial coronary artery stenosis [18] or microvascular disease in adults [19], but this has not been demonstrated in children.

The purpose of our study was therefore to (1) report our experience with the safety and feasibility of using regadenoson for myocardial stress perfusion MRI in a pediatric population, (2) evaluate the impact of MOCO on stress perfusion image quality, and (3) use semi-quantitative analysis to compare differences in myocardial perfusion reserve in patients with and without qualitative perfusion defects.

## 2. Materials and Methods

This HIPAA compliant retrospective study was approved by the Institutional Review Board (IRB) at Ann and Robert H. Lurie Children’s Hospital of Chicago on 7 June 2016 (IRB 2016-544). Consent for this study was waived per IRB protocol.

### 2.1. Study Population

All consecutive pediatric patients (age 0–21 years), who underwent a clinically indicated stress perfusion MRI with regadenoson from December 2014 to September 2017 were included in this retrospective review. Demographic data including gender, age, height, weight, and body surface area were recorded. Clinical history including original diagnosis, any surgical procedures performed and specific indication for the stress perfusion MRI examination were documented.

### 2.2. CMR Protocol

Patients were advised to refrain from the use of caffeine containing products for 24 h prior to the study. Informed consent for the clinically indicated stress MRI examination was obtained on the day of the study. Informed consent included a detailed review of the procedure and potential side effects of regadenoson (Lexiscan; Astellas Pharm; Northbrook, IL, USA). A pediatric cardiologist, pediatric radiologist, cardiac MR technologist and a pediatric nurse were present for the study. Anesthesia was provided as needed per clinical protocol. A cardiac resuscitation cart was present outside the MRI scan room.

Studies were performed on a 1.5-Tesla (T) clinical MRI scanner (Siemens Aera, Erlangen, Germany). Figure 1 outlines the sequences for the comprehensive stress perfusion MRI protocol followed in our MRI lab and previously published by our group [20].

In a standard clinical manner, image localizer scans, 2D cine bSSFP images in two- chamber, three-chamber and four-chamber planes were obtained (TR = 3.0 ms TE = 1.26–1.3 ms; flip angle = 90°, slice thickness = 6 mm, in plane resolution = 1.0 × 1.0 mm^2^). T2 mapping was performed followed by contrast injection. Three T2-prepared bSSFP images with T2-prep times of 0, 24, and 55 ms were acquired in a breath-held fashion in 3 short axis orientations—base (at the mitral valve leaflet tips), mid (between base and apex), and apex (apical ventricular cavity still visible at end-systole) through the left ventricle (TR = 2.5 ms, TE = 1.1 ms, slice thickness = 8 mm, in-plane resolution 1.9 × 1.9 mm^2^). Next, T1 mapping with a MOLLI sequence was used to measure native (pre-contrast) longitudinal relaxation T1 times of myocardium at the same three levels as well as within the blood pool. Pre-contrast T1 maps were obtained using a single-breath-hold, ECG-triggered, MOLLI sequence (TR = 2.6 ms, TE = 1.0 ms, slice thickness = 6 mm, in-plane resolution 0.7 × 0.7 mm^2^). Regadenoson was used to perform myocardial stress perfusion at a dose of 6–10 mcg/kg (up to a maximum dose of 400 mcg) [14] while closely monitoring the patient’s symptoms and vital signs. Following this, first-pass contrast-enhanced images were acquired at 60–90 s at the same three short axis levels. A gadolinium-based contrast agent was given intravenously at a dose of 0.05 mmol/kg and then single-shot TurboFLASH, advanced motion corrected (MOCO) fast gradient echo sequences were acquired during stress (TR = 2.5 ms TE = 1.1 ms, inversion time (TI) 150 ms, flip angle = 12°, slice thickness = 8 mm, inplane resolution = 2.8 × 2.8 mm^2^). Once the required images were acquired, a single dose of 75 mg of intravenous aminophylline was given to reverse the effects of regadenoson. In the duration between stress and rest perfusion, ECG-gated 2-D cine bSSFP imaging of the ventricles was obtained in short-axis from base to apex. A second dose of 0.05 mmol/kg dose of a gadolinium-based contrast agent was given and contrast-enhanced first pass perfusion imaging at rest was obtained. Then, the final, third dose of 0.05 mmol/kg of a gadolinium-based contrast agent was injected with post-contrast T1 mapping. Segmented inversion–recovery sequences in three orientations: 4-chamber, 2-chamber, and short axis from base to apex (TR = 2.8 ms, TE = 1.2 ms, slice thickness = 8 mm, flip angle 50°, inplane resolution = 1.4 × 1.4 mm^2^) were used to obtain LGE images between 20–30 min from the time of the initial contrast injection. For LGE sequences, the inversion time was selected using an inversion time scout scan to optimally null the normal myocardium. The various contrast agents administered over the duration of the study included gadopentate dimeglumine (Magnevist, Bayer, Whippany, NJ, USA), gadobenate diglumine (MultiHance, Bracco Diagnostics Inc, Monroe City, NJ, USA), gadoterate meglumine (Dotarem, Guerbet, Raleigh, NC, USA, or gadobutrol (Gadavist, Bayer, Whippany, NJ, USA) at 3 mL/s.

### 2.3. Stress Response Data

Vital signs were evaluated upon arrival, prior to scanning, and immediately prior to administration of regadenoson. Heart rate and blood pressure were documented periodically for an hour following regadenoson (every minute for 5 min, every 5 min for the next 20 min and then every 10–15 min until an hour after regadenoson administration). The time from administration of regadenoson to peak heart rate and the time required to return to baseline heart rate were documented. Patients were monitored for an hour after the completion of the study prior to discharge.

Adverse events and need for termination of the study were documented. Arrhythmia, hypotension, atrioventricular block, and bronchoconstriction were considered major adverse events. Tingling in the limbs, gastrointestinal effects such as nausea and abdominal pain, anxiety, and chest pain were considered minor adverse events.

### 2.4. Image Analysis

#### 2.4.1. Qualitative Myocardial Perfusion Analysis

Images from each examination were reviewed in consensus by 2 experienced cardiovascular imagers (a pediatric radiologist with 20 years of experience and a pediatric cardiologist with 12 years of experience in cardiac MRI), who were blinded to the clinical report impression. Qualitative analysis was performed based on consensus recommendations developed by the Task Force for Post Processing of the Society for Cardiovascular MR (SCMR) [21,22]. Both readers evaluated the overall image quality of the myocardial first-pass contrast-enhanced MRI perfusion short axis images acquired at stress and at rest. Image quality was assessed for both the native and MOCO reconstructions. Image quality was graded on a 5-point Likert scale with 1 = non-diagnostic, 2 = poor image but diagnostic, 3 = adequate, 4 = good, and 5 = excellent. The perfusion images were then assessed for perfusion defect defined as decreased signal intensity within the myocardium in either a subendocardial or transmural (greater than 25% wall thickness pattern). The presence of a defect at both rest and stress, with no LGE, was considered an artifact. If the defect was present both at rest and stress, and late gadolinium enhancement (LGE) was seen in the same myocardial region, it was described as fixed perfusion defect. However, if a defect was present at stress and not at rest, it was described as an inducible, reversible perfusion deficit (Figure 2b). The presence of associated regional wall motion abnormalities was used as an additional tool to aid in the determination of presence of an inducible perfusion defect. The presence of dark rim artifact, defined as a transient rim of subendocardial hypo intensity, seen at rest and stress, was recorded if present [21,22].

#### 2.4.2. Semi-Quantitative Myocardial Perfusion Analysis

Semi-quantitative myocardial perfusion analysis (Figure 3) was performed by a pediatric cardiologist with 3 years of experience in cardiac MRI who was blinded to the qualitative analysis using the commercially available software Medis 3.0 (Leiden, The Netherlands). The endocardial and epicardial borders were manually traced on all frames of rest and stress MOCO images, excluding the papillary muscles. A region of interest placed in the left ventricle served as the myocardial blood pool signal. Segmental time–signal intensity (TSI) curves were generated for the basal, mid, and apical slices during stress and rest states, and for the blood pool as shown in Figure 3a. Segmentation was based on the AHA 16 segment model. Variables were obtained from each time–signal intensity curve for each segment at stress and rest as shown in Figure 3b. The maximum rate of signal intensity increase per unit time (au/s) was defined as the maximum upslope. The maximum upslope of the segment TSI curve divided by the maximum upslope of the myocardial blood pool TSI curve was the relative upslope. The myocardial perfusion reserve index (MPRI) was then calculated as the ratio of the stress and rest relative upslopes for each segment. A global MPRI was also calculated as the mean MPRI of all segments for each patient. See Figure 4. Intraobserver and interobserver variability of the MPRI was calculated for a subset of patients (n = 10 and n = 9 for intraobserver and interobserver variability, respectively, approximately 25% of the cohort) by two pediatric cardiologists with 3 and 5 years of experience in cardiac MRI, respectively.

#### 2.4.3. Quantitative Measures of Cardiac Volume, Function, and Fibrosis

The short-axis cine steady-state free precession images of the left ventricle were processed to obtain the left ventricular volumes at end diastole and end systole (Medis 3.0, Leiden, The Netherlands). The left ventricular end diastolic volume indexed to body surface area (LVEDVi) and the left ventricular ejection fraction (LVEF) were calculated. Pre and post contrast T1 images were contoured and ECV was calculated using standard methodology [23].

#### 2.4.4. Statistical Analysis

Categorical variables are expressed as numbers and percentages. Continuous variables are presented as means ± standard deviations or median (with interquartile range–IQR) as appropriate. To assess significant group differences, an unpaired *t*-test (for normally distributed data) or a Wilcoxon rank-sum test (for non-normally distributed data) was performed between cases and controls. Bland–Altman plots were created to compare interobserver and intraobserver variability. Significance was determined by *p* < 0.05.

## 3. Results

### 3.1. Study Population

Thirty-nine consecutive pediatric patients, (23 males and 16 females) underwent a clinically indicated myocardial stress perfusion MRI with regadenoson from December 2014 to September 2017. One patient’s study was terminated prior to completion due to emesis in the MRI room after regadenoson infusion, and thus 38 patients completed the protocol. None of the patients underwent general anesthesia. The median age at the time of the study was 15 years (IQR 12–17) with a median weight of 61 kg (IQR 51–75), median height of 163 cm (IQR 154–170), with a body surface area of 1.67 m^2^ (IQR 1.48–1.83). The clinical diagnoses are listed in Table 1.

### 3.2. CMR Protocol Data

Of the 38 patients, 12 received a weight-based dose of 6 mcg/kg of regadenoson, one received a dose of 10 mcg/kg of regadenoson and the remaining (n = 25) received the maximum adult dose of 0.4 mg. The average time between end of administration of regadenoson to administration of aminophylline was 3 ± 1 min.

#### 3.2.1. Hemodynamic Response to Regadenoson

After administration of regadenoson, the average baseline heart rate rose from 86 ± 17 beats per minute (bpm) to 118 ± 18 bpm, showing a 41 ± 27% rise in heart rate (*p* < 0.005). There was no difference in the rise in heart rate for the patients who received weight-based dosing (6 mcg/kg) as compared to the patients who received a standard dose of 0.4 mg (*p* = 0.7). Peak heart rate occurred at 2 ± 1 min after administration of regadenoson and returned to baseline on an average 23 ± 19 min after peak heart rate. There was one patient who continued to have a heart rate higher than baseline 60 min after regadenoson but had no associated symptoms. The average systolic blood pressure was 109 ± 12 mm Hg and decreased to 98 ± 11 mm Hg (*p* < 0.005), at 6 ± 7 min after administration of regadenoson. The average diastolic blood pressure was 65 ± 11 mm Hg and decreased to 47 ± 9 mm Hg (*p* < 0.005), at 9 ± 7 min after administration of regadenoson. No patients needed in-hospital admission for prolonged monitoring.

#### 3.2.2. Adverse Events of Regadenoson

No patients had any major adverse events. One patient had emesis in the MR suite after administration of regadenoson, resulting in termination of the study. Of the remaining 38 patients, 15 (39%) had transient secondary side effects such as limb tingling (n = 4), nausea/gastrointestinal discomfort (n = 3), anxiety (n = 3), chest pain (n = 3), and headache (n = 2). All 15 had complete resolution of symptoms prior to the end of the exam.

### 3.3. Image Analysis

#### 3.3.1. Qualitative Myocardial Perfusion Analysis

On a 5-point Likert scale, 100% of the studies were of adequate or better image quality as reported in consensus by two experienced cardiovascular imagers. The image quality of MOCO reconstructions was significantly better than native reconstructions, both at rest (4.63 vs. 4.01, *p* < 0.005) and during stress (4.41 vs. 3.84, *p* < 0.005). The presence of a dark rim artifact was frequent during stress, both in the MOCO and non MOCO images (71% and 65%, respectively).

Four of the 38 patients had a reversible perfusion defect on the first pass perfusion images. Clinical details of these four patients are described in Table 2. None of them had undergone an orthotopic heart transplant. None had a defect at rest. One of these four patients was noted to have wall motion abnormalities in the area of the perfusion defect. Of the remaining 34 patients, two other patients were noted to have late gadolinium enhancement (one with history of Kawasaki disease and another who had D-TGA s/p arterial switch operation).

#### 3.3.2. Semi-Quantitative Myocardial Perfusion Analysis

The global and segmental MPRI’s are reported in Table 3 below.

The global MPRI of the 16 patients who had undergone orthotopic heart transplant was significantly lower than the non-transplant patients (0.75 ± 0.22 vs. 0.92 ± 0.23, *p* = 0.03). The global MPRI of the four patients with reversible perfusion deficits did not differ from the patients without perfusion defects. The segmental MPRI in the four patients with reversible perfusion defects, using segments that were noted to be abnormal on the qualitative assessment, were compared to the mid septal MPRI of the non-transplant patients, and were noted to be lower nearing statistical significance 0.78 ± 0.22 vs. 0.99 ± 0.36, *p* = 0.07).

There was no significant interobserver (n = 9, r = 0.84 with 95% CI of 0.47 to 0.96, *p* < 0.001) or intraobserver (n = 10, r = 0.84, with 95% CI of 0.51 to 0.96, *p* < 0.001) variability in the measurement of the global MPRI as determined by assessing the intraclass correlation. Bland–Altman plots were created to depict the inter- and intra-observer differences in the global mean MPRI. (See Figure 5). Overall, the plots showed that the differences between the interobserver and intraobserver measurements of MPRI were small with a good coefficient of variation (12.7% and 10%, respectively).

#### 3.3.3. Quantitative Measures of Cardiac Volume, Function, and Fibrosis

There was no significant difference in LVEDVi between the patients with perfusion deficits and those without perfusion defects (85 mL/m^2^ vs. 91 mL/m^2^, *p* = 0.32). The mean LVEF was normal in both groups although higher in those with perfusion deficits (65% vs. 60%, *p* = 0.01). The pre-contrast global T1 values were significantly lower in those with perfusion deficits as compared to those without, but still within normal range for both groups (999 ± 34 vs. 1031 ± 34, *p* = 0.02). There was no significant difference in the post contrast global T1 values (520 ± 42 vs. 492 ± 67, *p* = 0.49) and ECV (24% ± 4% vs. 25% ± 4%, *p* = 0.34) in patients with and without perfusion deficits.

### 3.4. Clinical Course

The clinical course of the four patients with the reversible perfusion deficits is described in Table 4. The indications for and results of the stress perfusion test and investigations performed after the stress perfusion study are described in this table. Two patients had known coronary artery stenoses, one was diagnosed with a hemodynamically significant myocardial bridge, and one had a coronary artery course that was acutely angulated and compressed.

## 4. Discussion

### 4.1. Safety and Efficacy of Regadenoson

In our experience, regadenoson is a safe and effective agent to induce myocardial pharmacological stress in children undergoing stress perfusion MRI for various indications. Our study showed that regadenoson did not have any major adverse side effects such as arrhythmias, hypotension, atrioventricular block, or bronchoconstriction. The only patient in whom the study was terminated, was secondary to anxiety followed by an episode of emesis. Overall, our study did have a somewhat higher incidence of minor events as compared to prior pediatric studies [13], although similar to adult studies [11,24,25]. As in our study, these minor events have been noted to be short-lived, benign, and to spontaneously terminate. This suggests that some physiological side effects of regadenoson are to be expected, requiring close monitoring of the patient during the study and appropriate counseling may improve the overall tolerance to these expected side effects.

In the adult population [26], and a small pediatric population [13,15], regadenoson has been shown to be an effective selective vasodilator with better side-effect profile as compared to adenosine. Moreover, the use of single bolus dose of regadenoson via a single intravenous access line significantly improves the ease of administering regadenoson compared to adenosine, especially in the pediatric population. The pharmacokinetics and safety of regadenoson has been previously studied and shown that, in adults, the maximum tolerated dose is 20 mcg/kg in the supine position. We used a weight-based dose of regadenoson, starting primarily at 6 mcg/kg, with a maximum dose of 400 mcg and showed an expected hemodynamic response with this dosing, with no difference between patients who received weight based versus standard dosing. Our study population was younger than previously studied pediatric patients, where a dose of 10 mcg/kg had been used [13]. Our study shows feasibility in a larger population with an overall lower weight and age range than previously reported.

Regadenoson, with its selective adenosine A2 receptor target activity makes it an ideal choice to diagnose myocardial ischemia in children with suspected fixed coronary artery obstruction, allowing for maximal coronary vasodilation. Although there is no direct evidence of maximal coronary vasodilation with regadenoson, the significant rise in heart rate that was demonstrated in this study and in prior investigations [13,26] shows that there is a vasoactive impact of regadenoson. As shown in our study and others [27], achievement of a peak heart rate within two minutes of administration, can aid in the assessment of perfusion deficits and regional wall motion abnormalities in a relatively short duration of time.

Stress MRI has been shown to have high specificity and sensitivity for detection of coronary artery disease in large adult trials [4,5]. In addition, MRI can evaluate complex cardiac anatomical structure, myocardial structure, and function on the same test without the potentially harmful effects of ionizing radiation [6]. These advantages are further heightened in the pediatric age group where faster heart rates, complex anatomy, and artifact from motion and breathing tend to complicate image acquisition. MOCO allows for improved qualitative assessment of perfusion defects as was seen with improved image quality scores noted for both stress and rest images. Further, this improvement in image quality facilitates semi-quantitative assessment of myocardial perfusion.

### 4.2. Semiquantitative Assessment of Myocardial Perfusion Reserve

Research in the quantitative analysis of myocardial perfusion has been driven by the desire to obtain observer independent and reproducible measures of myocardial perfusion. Its use in the clinical realm is restricted by the time-consuming operator interaction needed for analysis. New automated software can significantly decrease the amount of time needed by the operator thereby bringing us closer to incorporating quantitative analysis into clinical use [28]. Semi-quantitative analysis examines the signal intensity within a myocardial region of interest over time generating a time–signal intensity (TSI) curve that can be used to derive parameters such as peak signal intensity, relative upslope, time to peak, mean transit time, area under TSI curve, and myocardial perfusion reserve index (MPRI). Among these various parameters the upslope and MPRI have been considered the most relevant semi-quantitative parameters for detection of a perfusion abnormality [10,29]. Our study used semi-quantitative assessment of myocardial perfusion in the pediatric population using regadenoson as the pharmacological agent. About 40% of our patient population were patients who had undergone an orthotopic heart transplant and given the known association of cardiac allograft vasculopathy and reduced global MPRI in these patients [20], we separated this subset of patients to make more meaningful comparisons. With no significant interobserver or intraobserver variability in the measurement of the global MPRI, out study showed that the global MPRI was lower in patients who had undergone an orthotopic heart transplant than the non-transplant patients. In the patients who had a reversible perfusion deficit, the segmental MPRI of the affected segments was lower than the mid septal MPRI of the non-transplant patients. A similar study performed in the adult population compared the global mid MPRI in patients who had undergone an orthotopic heart transplant with normal controls and patients with a history of myocardial infarction with MPRI of 0.5, 0.74, and 0.77 in the three groups, respectively [30]. Moreover, patients with no or mild coronary allograft vasculopathy had a higher MPRI compared to those with more significant coronary allograft vasculopathy [30]. Although there are no data that establish normal MPRI in the pediatric population, the lower segmental MPRI in the group with perfusion deficits suggests that MPRI might be used as an adjunctive tool to detect a myocardial perfusion abnormality. Further studies in larger cohorts will be needed to determine clinical utility. In the pediatric population, stress perfusion MRI is useful in a broad array of disease processes, including patients who have undergone orthotopic heart transplant, have coronary artery pathology secondary to Kawasaki disease, or have undergone surgical manipulation of their coronaries based on their underlying congenital heart defect or have anomalous origins of the coronary arteries [13,31]. Studies have also shown that in patients with hypertrophic cardiomyopathy, with histopathologic changes in myofibrillar disarray and abnormal intramural coronary vasculature, there is reduced resting myocardial perfusion [32]. The use of semi-quantitative assessment of myocardial perfusion as an adjunct to qualitative assessment of myocardial perfusion is useful not only in the interpretation of localized ischemia but could be especially useful in the setting of diffuse ischemia secondary to microvascular dysfunction [19]. Our study suggests that the use of myocardial perfusion reserve index based on a semiquantitative analysis can be used as an additional corroborating tool to assess for the lack of adequate myocardial perfusion in children.

## 5. Conclusions

Pediatric myocardial stress perfusion can be performed safely using regadenoson, in a closely monitored setting. The utility of MOCO imaging improves the image quality for qualitative analysis of perfusion deficits. Semi-quantitative assessment of myocardial perfusion is feasible and reliable. It can be used as an adjunct tool in determining the presence of inducible myocardial perfusion deficits. Larger pediatric studies evaluating the association of myocardial perfusion reserve with clinical outcomes are warranted.

## Figures and Tables

**Figure 1 children-09-01332-f001:**
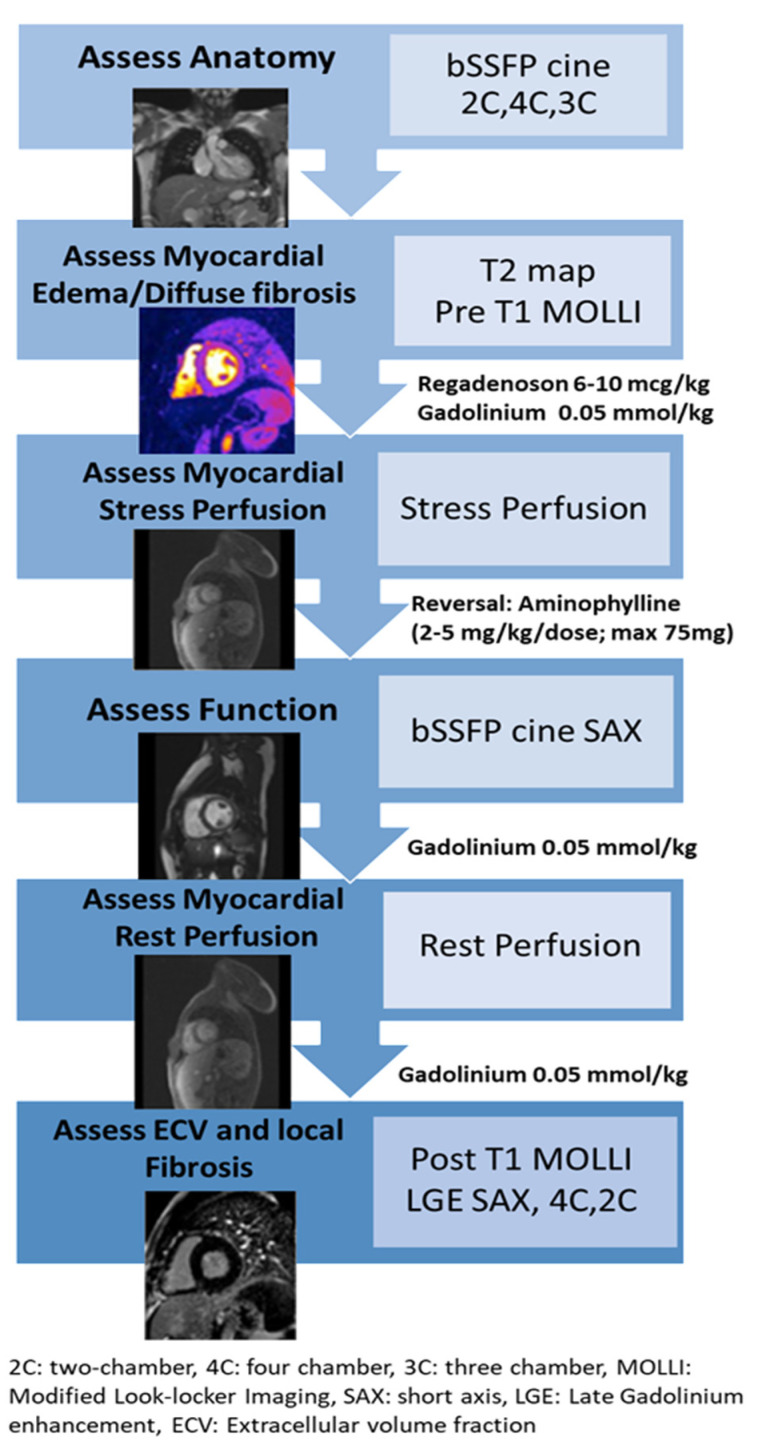
Stress Perfusion MRI Protocol.

**Figure 2 children-09-01332-f002:**
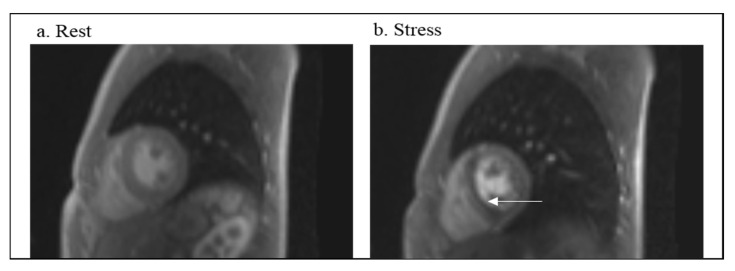
Qualitative assessment of myocardial perfusion deficits. (**a**) demonstrates homogenous myocardial contrast enhancement in the rest state. (**b**) demonstrates a mid- to submyocardial inducible perfusion defect (white arrow) during the stress state. This 17-year-old male underwent stress perfusion CMR after undergoing coronary unroofing with creation of a neo-ostium for anomalous left coronary artery arising from the right sinus of Valsalva.

**Figure 3 children-09-01332-f003:**
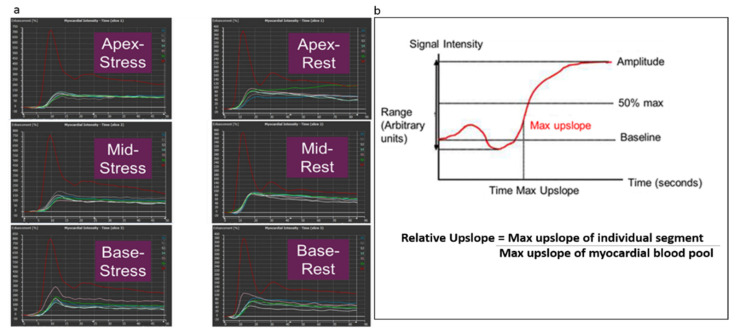
Semi-quantitative myocardial perfusion analysis. (**a**) shows time–signal intensity (TSI) curves for basal, mid ventricular and apical slices during rest and stress states. The TSI curve in red represents the myocardial blood pool. The other curves of various colors represent the myocardial segments at the basal, mid ventricular and apical levels. (**b**) demonstrates all the variables obtained from each time–signal intensity curve for each segment at stress and rest state, and calculation of the relative upslope as maximum upslope of individual segment/maximum upslope of myocardial blood pool.

**Figure 4 children-09-01332-f004:**
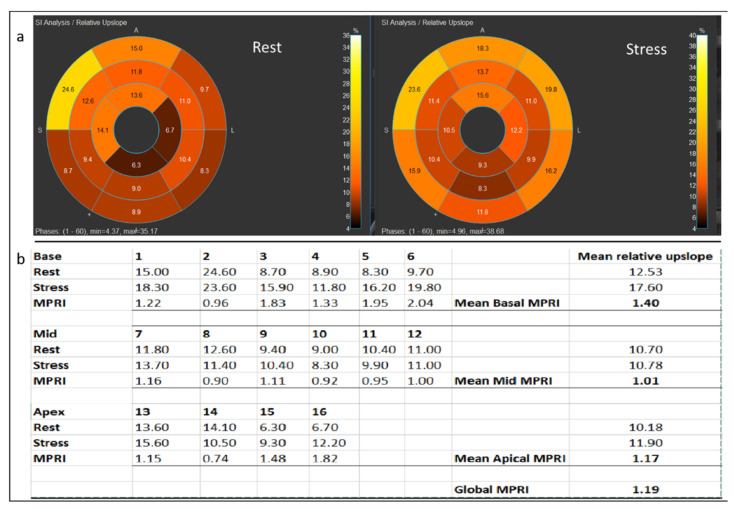
Calculation of global MPRI using relative upslopes at rest and stress. (**a**) shows the AHA 16 segment model depicting the relative upslope at rest and stress in the same 17-year-old male who underwent stress perfusion MRI after coronary unroofing with creation of a neo-ostium for anomalous left coronary artery arising from the right sinus of Valsalva. (**b**) shows how MPRI was calculated for each of the 16 segments as a ratio of the relative upslope at stress to relative upslope at rest. Basal, mid, and apical MPRI was also calculated. Global MPRI is the mean MPRI of all segments.

**Figure 5 children-09-01332-f005:**
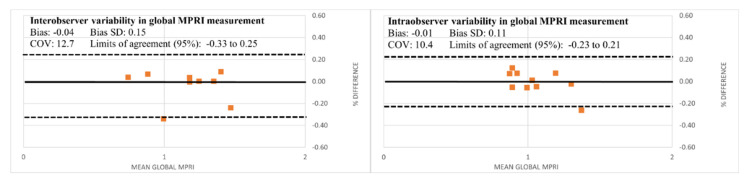
Bland–Altman plots for intraobserver and interobserver variability in measuring the mean global MPRI, with coefficient of variation (COV), bias and limits of agreement.

**Table 1 children-09-01332-t001:** Clinical diagnosis of 38 patients undergoing stress perfusion MRI.

Clinical Diagnosis	Number
Status post orthotopic heart transplant	16
Coronary aneurysms	10
- Isolated coronary aneurysm with mitral regurgitation	- 1
Status post coronary artery revision	8
- D-TGA status post arterial switch	- 3
- Anomalous RCA from left sinus s/p unroofing	- 3
- Anomalous LCA from right sinus s/p unroofing and creation of a neo-ostium	- 1
- Congenital aortic stenosis s/p Ross	- 1
Abnormal coronary artery anatomy, no intervention	3
- Single coronary artery with intramural LAD	- 1
- Single coronary artery with LAD between aorta and PA	- 1
- Anomalous LCA from non-coronary sinus	- 1
Anginal chest pain with abnormal ECG	1

D-TGA D-loop transposition of the great arteries, RCA—right coronary artery, LCA—left coronary artery, LAD—left anterior descending coronary artery, PA—pulmonary artery, ECG—electrocardiogram.

**Table 2 children-09-01332-t002:** Clinical diagnosis of 4 patients with reversible perfusion deficits.

Patient	Diagnosis	Stress Perfusion Defect	Rest Perfusion Defect	Wall Motion Abnormality	Late Gadolinium Enhancement
1	Kawasaki	Yes	No	No	No
2	DTGA s/p ASO	Yes	No	Yes	No
3	AAOLCA from right sinus s/p surgery	Yes	No	No	No
4	Kawasaki	Yes	No	No	No

**Table 3 children-09-01332-t003:** Comparison of global and segmental MPRI in patients with and without perfusion defects.

	Patients with Perfusion Defect (n = 4)	Patients without Perfusion Defect s/p Orthotopic Heart Transplant (n = 16)	Patients without Perfusion Defect Non-Transplant (n = 18)	
Global MPRI	1.13 ± 0.27	0.75 ± 0.22	0.92 ± 0.23	*p* = 0.03 *
Segmental MPRI	0.78 ± 0.22		0.99 ± 0.36	*p* = 0.07

* Lower global MPRI in patient’s s/p heart transplant vs. non-transplant patients.

**Table 4 children-09-01332-t004:** Clinical follow up of 4 patients with reversible perfusion deficits.

Patients with Stress Perfusion Deficits			Tests Performed Post Perfusion				
	Diagnosis	Indication	CTA	Cath	NM Stress	EST	Outcome
1	AAOLCA from right sinus s/p surgery	exertional chest pain	No	* LAD bridge	Normal	Normal	Surgical myocardial bridge unroofing
2	Kawasaki	Coronary aneurysm with LAD occlusion	No	No	Normal	Normal	No intervention
3	D-TGA s/p ASO	Acutely angulated and compressed reimplanted LMCA	No	No		Normal	Started on aspirin
4	Kawasaki	Coronary aneurysm with LAD occlusion	No	No	Normal	Normal	No intervention

* Hemodynamically significant LAD bridge. AAOLCA—Anomalous aortic origin of left coronary artery, ASO—arterial switch operation, CTA-Computed Tomography Coronary Angiogram, Cath- Catheter based coronary angiogram, D-TGA D-loop transposition of the great arteries, EST—Exercise stress test, LAD—left anterior descending coronary, LMCA—left main coronary artery, NM stress- nuclear medicine scintigraphy with stress.

## Data Availability

Not applicable.

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
