# Peer review of "Safety and Efficacy of Regadenoson for Pediatric Stress Perfusion Cardiac MRI with Quantification of Myocardial Blood Flow"

_children, 2022, doi:10.3390/children9091332_

Round 1
Reviewer 1 Report
This is a very well written and clinically useful study on the use of regadenoson for pediatric stress perfusion cardiac MRI. I think the authors have a sound study design, analysis plan and make appropriate and meaningful conclusions. The conclusions have both practical utility for pediatric cardiologists, especially imaging specialists, and the safety and image quality data are useful for any pediatric cardiologist in practice as CMR becomes increasingly utilized.
I have no major concerns or recommendations for the authors. I believe this article merits publication and I congratulate the team on a great project.
There are a few minor comments I would like to make:
- The abbreviation ECV is used twice in the paper but I don't believe there is any expansion of this abbreviation so I was not certain what ECV stands for.
- Figure 5 is a little tough to read due to the size of the text, but I expect this will be revised in future steps of this process.
- I found Table 4 to be a little bit confusing with the "No" for CTA and cath. Does "no" mean the patients did not undergo those tests? Or that the patient had no findings? There is a blank entry for Patient 3 under NM stress which should be filled in. Perhaps some abbreviation for not performed or not applicable may be more clear than "no"?
Author Response
"Please see the attachment"

Reviewer 2 Report
This is an excellent paper written by Patel et al. titled " Safety and efficacy of Regadenoson for pediatric stress perfusion cardiac MRI with quantification of myocardial blood flow". The authors have described the use of this medication in pediatric population. This is a heterogeneous group of patients and are well represented in the study.
Here are a few clarifications:
- Aminophylline dose was used in few patients based on weight and some of them were given the adult dose, however, there was no significant side effect from aminophylline, hence, to clarify, are the authors suggesting using a standard adult dose for pediatric population as well?
- Most of the patients were over 10y old, however, we do notice perfusion defects in children lower than that and after 1-2 years post operative or post-transplant period: Why were patients below 10 y not selected? In addition, is there any effect of anesthesia if any of the patients had undergone general anesthesia for the study?
Minor comment/correction:
- Page2 - para3.64 - allow improved semi-quantitative and quantitative assessment of myocardial perfusion.... To clarify, I am assuming you meant semi-quantitative and qualitative....
Author Response
"Please see the attachment"
